# Long distance multiplexed quantum teleportation from a telecom photon to a solid-state qubit

Dario Lago-Rivera [1] ✉, Jelena V. Rakonjac[1], Samuele Grandi [1] & Hugues de Riedmatten [1,2] ✉

Quantum teleportation is an essential capability for quantum networks, allowing the transmission of quantum bits (qubits) without a direct exchange of quantum information. Its implementation between distant parties requires teleportation of the quantum information to matter qubits that store it for long enough to allow users to perform further processing. Here we demonstrate long distance quantum teleportation from a photonic qubit at telecom wavelength to a matter qubit, stored as a collective excitation in a solid-state quantum memory. Our system encompasses an active feed-forward scheme, implementing a conditional phase shift on the qubit retrieved from the memory, as required by the protocol. Moreover, our approach is time-multiplexed, allowing for an increase in the teleportation rate, and is directly compatible with the deployed telecommunication networks, two key features for its scalability and practical implementation, that will play a pivotal role in the development of long-distance quantum communication.

Transferring quantum information between remote parties is a basic and still challenging requirement in the field of quantum communication. Quantum teleportation was first proposed in 1993[1] and demonstrated a few years later[2–4] using light. This protocol allows the transfer of quantum states using previously-shared entanglement and classical communication. The qubit to be teleported is measured jointly with one part of the entangled state, projecting it onto a Bell state. It is of fundamental interest to have access to the qubit after the teleportation takes place as it allows for further use of the quantum information once it has been transferred. Besides, the teleportation protocol requires the application of a unitary transformation to the stored qubit based on the result of the remote Bell-state measurement (BSM)[4–6]. To achieve this goal over long distances, the desired qubit should be teleported to a matter qubit stored in a quantum memory featuring a storage time longer than the two-way communication time between parties. Moreover, the required entanglement transfer over several km of quantum communication channels would benefit from compatibility with the already deployed telecommunication infrastructure. In addition, multiplexed operation is highly desirable[7] to allow for a favourable scaling with respect to schemes with single mode memories, which would have to wait for the whole communication time before executing a new teleportation attempt. Quantum teleportation of arbitrary quantum states has been demonstrated with various physical systems and in different implementations[2–6,8–16]. Entanglement between spatially separated quantum material systems has also been demonstrated using entanglement swapping[17–19] including a long distance demonstration with two NV-centers separated by 1.6 km[20], and recently with two single atoms separated by 33 km of optical fibres and telecom heralding[21]. Multiplexed quantum teleportation of arbitrary states to a matter qubit over a long distance has however never been shown.

Rare earth-doped crystals are promising candidates for storing qubits. They provide a compact platform composed of a large number of atoms naturally trapped in a crystalline structure. At cryogenic temperatures they feature excellent coherence properties and moreover they allow for multiplexing in several degrees of freedom[22–24], a significant advantage with respect to other systems[7,18]. The use of rare-earth doped crystals has lead to demonstrations of storage[25–27] and

[1]ICFO-Institut de Ciencies Fotoniques, The Barcelona Institute of Science and Technology, 08860 Castelldefels (Barcelona), Spain. [2]ICREA-Institució Catalana de Recerca i Estudis Avançats, 08015 Barcelona, Spain. ✉e-mail: dario.lago@icfo.eu; hugues.deriedmatten@icfo.eu

generation[28–31] of quantum states of light. They have also been involved in quantum teleportation experiments, which however featured short distances[18,19] or storage times short compared to the communication time[13].

Here we report multiplexed quantum teleportation from a photonic qubit at telecom wavelength to a matter qubit separated by 1 km of optical fiber. The qubit is encoded as a collective excitation in a solid-state quantum memory based on a praseodymium-doped crystal and is stored for a duration longer than the two-way communication time. Thanks to the multimodality featured in our architecture we were able to increase the repetition rate of our experiment by a factor of about three beyond the limit set for a single mode equivalent without degrading the fidelity of the teleported state. Moreover, as we still have access to the qubit after processing the result of the remote BSM, we are able to perform a unitary transformation and therefore to add active feed-forward control as required by the teleportation protocol.

## Results

### Experimental set-up

The experiment consists of two parts which we refer to as Alice and Bob (Fig. 1a). At Alice there is a source of light-matter entanglement composed of an energy-time entangled photon pair source and of a quantum memory (QM) enabling the storage of qubits. The entanglement source is based on cavity-enhanced spontaneous parametric down conversion (cSPDC)[27,32], designed to emit 1.8 MHz-wide photons. One photon of the pair is at 606 nm and it is stored in a $Pr^{3+}$:$Y_2SiO_5$ crystal as a collective excitation of $Pr^{3+}$ ions using the Atomic Frequency Comb (AFC) protocol[33]. Here a comb-like absorption structure is prepared on the optical transition of the inhomogeneously broadened ions (Fig. 1b). Stored photons are re-emitted after a fixed time given by the inverse of the periodicity of the comb. The second photon of the pair is instead at telecom wavelength and is sent through an optical fibre towards Bob. At Bob there is a time-bin qubit source consisting of an amplitude and phase modulator (AOM in Fig. 1a) that shapes light generated by an optical parametric oscillator as two weak coherent pulses $|e\rangle$ and $|l\rangle$, separated by 420 ns and with an average number of photons $\mu \ll 1$. It is then possible to generate an arbitrary superposition state (Fig. 1c): $\alpha|e\rangle + e^{i\varphi}\beta|l\rangle$ with $\alpha^2 + \beta^2 = 1$ and $\varphi$ representing the phase relation between the two time-bins.

### Bell-state measurement

We implement the BSM for the teleportation protocol by sending the generated qubits to interfere with telecom idler photons from Alice at a beam splitter (BS) and by detecting the photons at the output modes with superconducting nanowire detectors (Fig. 2a). Provided that the modes at the BS are indistinguishable, two consecutive detections at the same output of the BS project the joint state of the telecom time-bin qubit and the idler photon into the Bell-state $|\Psi^+\rangle = 1/\sqrt{2}(|el\rangle + |le\rangle)$ that heralds the teleportation of the $e^{i\varphi}\beta|e\rangle + \alpha|l\rangle$ qubit into the $Pr^{3+}$ ions. On the other hand, two consecutive detections in the two time-bin modes in different BS outputs project the joint state into the Bell-state $|\Psi^-\rangle = 1/\sqrt{2}(|el\rangle - |le\rangle)$, heralding instead the teleportation of the $e^{i\varphi}\beta|e\rangle - \alpha|l\rangle$ qubit. We send the detection triggers back to Alice where we use an electronic logic control system in order to discriminate between these two scenarios. Using a $\pi$-phase shifter (PS) that we switch on conditioned on the detection of a $|\Psi^-\rangle$ event we make sure that the heralded state after the memory is always $e^{i\varphi}\beta|e\rangle + \alpha|l\rangle$. The additional bit flip required to recover the original qubit (not performed in this experiment) could be implemented using an unbalanced Mach-Zehnder interferometer (see Supplementary Note 2).

### Short distance quantum teleportation

In the first test, we separated Alice and Bob by a few meters and prepared the QM to store the signal photons for 10 μs, with a storage and retrieval efficiency $\eta_{AFC} = 18.8(5)$ %. We prepared the qubits $|e\rangle$, $|l\rangle$, $|+\rangle = 1/\sqrt{2}(|e\rangle + |l\rangle)$, $|R\rangle = 1/\sqrt{2}(|e\rangle + i|l\rangle)$, and we configured Bob to send qubits every 4.1 μs, i.e. with a repetition rate of 244 kHz. For every input qubit, we analysed using its parallel and orthogonal settings in order to estimate the fidelity of the state after the phase shifter (see Methods). To characterise the teleportation fidelity for states on the equator of the Bloch Sphere ($\bar{F}_{eq}$), we used a second $Pr^{3+}$:$Y_2SiO_5$ crystal (analysis crystal) where an AFC was prepared with a storage time of 420 ns. By balancing the storage efficiency such that the photons could be stored or transmitted through the memory with equal probability, the $|e\rangle$ and $|l\rangle$ components of the teleported qubit were overlapped in time and therefore interfered (Fig. 2b)[27]. To analyse the fidelity of the teleported pole states ($\bar{F}_{poles}$), we prepared a transparency window of 16 MHz in the analysis crystal and we compared the relevant time bins of the time-resolved coincidence histogram (see Supplementary Note 3 and ref. 8). The mean fidelity of an arbitrary qubit is

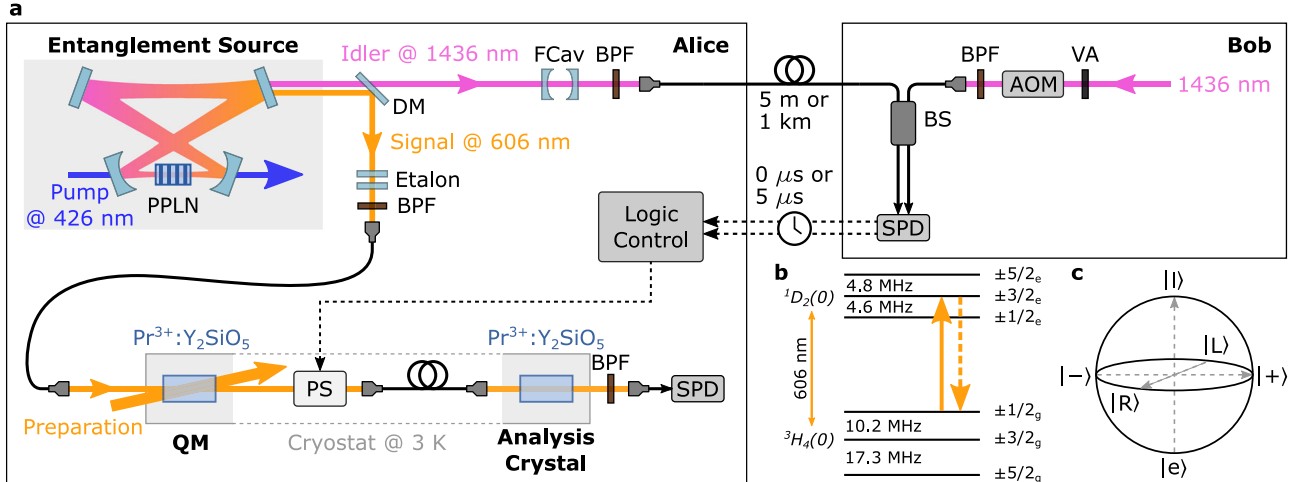

**Fig. 1 | Experimental set-up. a** Entangled photon pairs are generated at Alice. Signal photons are routed to a Pr-doped crystal, while the idler photons are sent towards Bob either through a 5 m or a 1 km long optical fibre. At Bob, arbitrary qubits at 1436 nm are produced and interfered with the idler photons in order to perform a BSM. Detection results are communicated back to Alice, where they are processed such that the feed-forward can be correctly applied. Periodically-polled lithium niobate (PPLN), dichroic mirror (DM), band pass filter (BPF), phase shifter (PS), single photon detector (SPD), filter cavity (FCav), acousto optical modulator (AOM), variable attenuator (VA). **b** Relevant level scheme of the $Pr^{3+}$:$Y_2SiO_5$ crystal. **c** Bloch-sphere where all the relevant qubit states for our experiment are labelled.

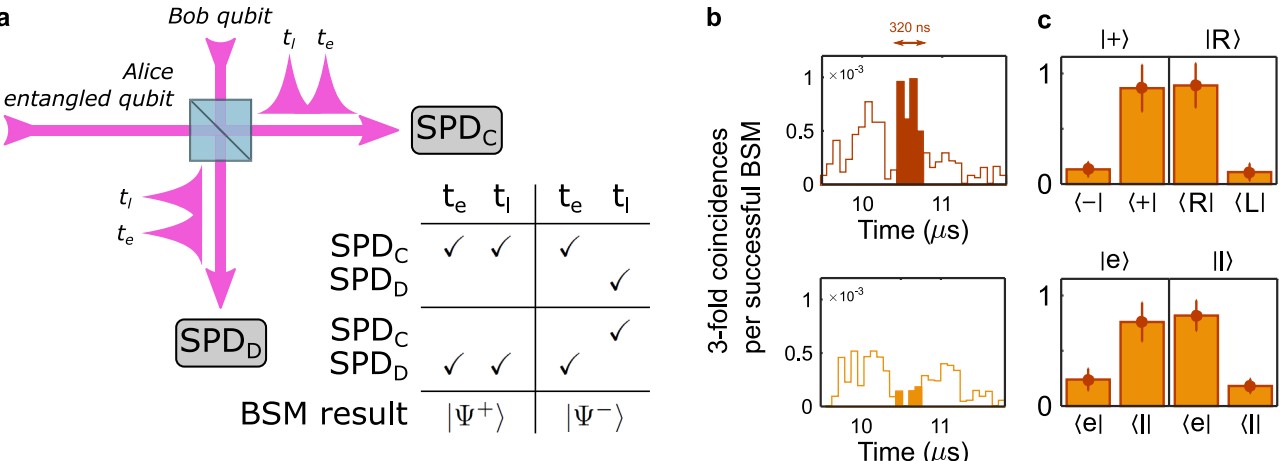

**Fig. 2 | Short distance quantum teleportation. a** Schematic of the working principle of the time-bin Bell-state measurement. Consecutive clicks at any detector project the input qubit-idler qubit joint state into $|\Psi^+\rangle$, while alternate detections project it into $|\Psi^-\rangle$. **b** 3-fold coincidence histograms after parallel and orthogonal analysers, renormalised to the total amount of successful BSMs for an input state $|R\rangle$. Three coincidence peaks appear in the analysis where only the central one corresponds to a projection in the superposition basis. **c** Normalised coincidences for different input qubits using a parallel and orthogonal setting of the analyser. The two upper plots correspond to the equator measurements and the two lower plots correspond to the measurements of the poles. The error bars correspond to $\pm 1\sigma$.

$\bar{F} = \frac{1}{3}\bar{F}_{poles} + \frac{2}{3}\bar{F}_{eq}$[13]. We measured $\bar{F}_{poles} = 80(5)\%$ and $\bar{F}_{eq} = 88(5)\%$ (Fig. 2c), resulting in $\bar{F} = 85(4)\%$, which is significantly above the 66.7% classical limit for single qubits[34] and the 73.6% limit considering the statistics of the weak coherent input qubits and the efficiency of the quantum channel (see Methods). The main reasons for the reduced fidelity are the limited fidelity of the shared entangled state, the finite indistinguishability between the entangled telecom photon and the qubit to be teleported and the fact that we encode the input qubits as weak coherent states (see Supplementary Note 3).

## Temporal multiplexing capabilities

If Alice and Bob are spatially separated, the teleportation repetition rate is limited by the two-way communication time between the two parties. For example, a separation of 1 km would correspond to a communication time of 10 μs, equal to the storage time of the QM. Thanks to the temporal multimodality of our quantum memory, we can overcome this limitation. We repeated the experiment varying the teleportation repetition rate from 133 kHz to 323 kHz, using $|R\rangle$ as the teleported qubit (Fig. 3a). We observed a constant fidelity for all the explored repetition rates (Fig. 3b). Note that the maximum rate used (323 kHz) was restricted by the speed of the experimental control electronics. Implementing a faster controller would allow the system to reach a repetition rate up to 1.19 MHz, only limited by the total duration of the time-bin qubit of 840 ns. If instead only a single qubit could be stored, after 1 km of distance the two-way communication time between Alice and Bob would lead to a maximum repetition rate of 100 kHz (black vertical line of Fig. 3b). The multimode operation then introduces a maximum gain of a factor of 12 with respect to a single mode architecture. This number corresponds to how many time-bin qubits can be stored in the quantum memory, equal to half the total number of temporal modes which can be stored (about 24 for this storage time). In combination with the telecom compatibility of our approach, the temporal multimodality of our QM makes our system especially suitable to be used in a long-distance scenario.

## Long distance quantum teleportation

We extended the distance between Alice and Bob and repeated the fidelity measurements with various teleported states (Fig. 3c). Alice's telecom photon had to travel through 1 km of optical fibre, taking 5 μs to reach Bob. An additional electronic delay of 5 μs was introduced after the detection events to simulate the communication time

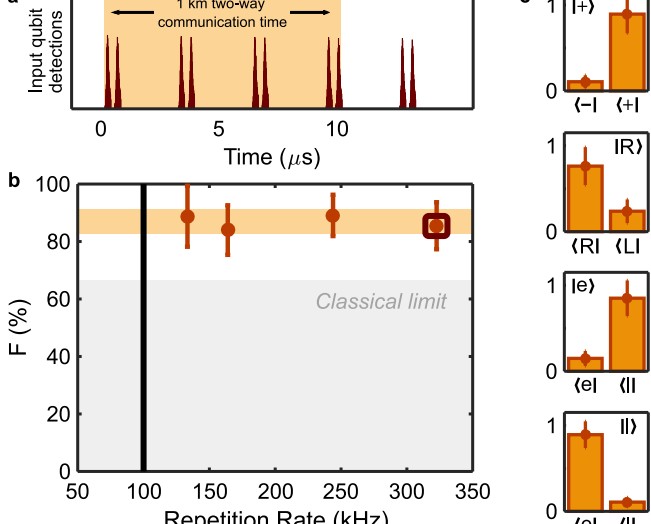

**Fig. 3 | Multiplexing capabilities and long distance teleportation. a** Example of the qubit sequence that we send from Bob. Each pair of peaks represents a qubit teleportation attempt. In this example, four teleported modes are stored at the same time (orange shaded area). **b** Fidelity versus teleportation repetition rate. The highlighted point corresponds to the qubit sequence of Fig. 3a. The light grey shaded area represents the classical limit and black line shows the maximum repetition rate that a single mode memory could have when used for teleportation over a 1 km distance. The solid orange band represents a $\pm 1\sigma$ variation with respect to the mean value between all the measured fidelities. **c** Long distance quantum teleportation. We used 1 km of optical fibre between Alice and Bob and an AFC in the QM of 17.5 μs. The histograms show the normalised coincidences for different input qubits using a parallel and orthogonal setting of the analyser. The error bars correspond to $\pm 1\sigma$.

between two locations separated by 1 km of optical fibre. Therefore, Alice had to wait a total of 10 μs to be able to process the BSM and act accordingly on the PS. During this time, Bob was unconditionally performing teleportation attempts every 4.1 μs. In order to keep the teleported qubit stored during this process we further extended the storage time of the QM to 17.5 μs, leading to $\eta_{AFC} = 12.2(4)\%$. We repeated the previous analysis of the teleportation protocol and

measured $\bar{F}_{poles}$ = 87(5) % and $\bar{F}_{eq}$ = 86(6) %, resulting in $\bar{F}$ = 86(4) %, which again violates the classical bounds by several standard deviations (see Methods). Note that by the time Alice receives the result of the remote BSM, the teleported qubit will still be stored in the QM for an additional 7.5 $\mu s$.

## Discussion

We have demonstrated here long-distance quantum teleportation of a photonic telecom qubit into a solid-state qubit with active feedforward. We applied a phase shift to the teleported qubit after the QM conditioned on the result of the BSM, such that the teleported qubit is always in the same state. Moreover, we used a temporally-multiplexed protocol which allowed us to increase the rate of teleportation attempts beyond the limit imposed by the communication time, without degrading the quality of the teleported qubit. For quantum teleportation over a distance of 1 km our approach resulted in a repetition rate three times higher than for the case of a single-mode memory, only limited by the speed of the logic hardware. This difference will become more pronounced for larger distances between Alice and Bob.

In order to further extend the distance between Alice and Bob we could increase the storage time of our matter qubits using the long-lived spin state of the $Pr^{3+}$ ions with dynamical decoupling techniques[29,35,36], which would also allow on-demand read-out of the stored qubits[27]. Moreover, higher rates of teleportation attempts can be achieved by combining the already existing temporal multimodality with other degrees of freedom already demonstrated in this system[22–24].

Our results show that qubits can be teleported in a multiplexed fashion onto a distant solid-state quantum memory and be further manipulated. They represent a functional and scalable realisation of long-distance quantum teleportation. This technique may also be used to transfer qubits between different type of quantum nodes in a hybrid quantum network. Future systems based on this architecture will enable remote quantum information distribution and processing.

## Methods

### Entanglement source

In order to generate photon pairs entangled in time we exploited the natural energy-time entanglement exhibited in SPDC processes if the coherence time $\tau_{pump}$ of the pump laser is much longer than the coherence time $\tau_{pair}$ of the photon pairs created. When this condition is fulfilled a coherent generation of one photon pair will be allowed during the coherence time of the pump beam. However, due to energy conservation, both photons of the pair will be generated in the same time bin. In the case of a cSPDC source, the coherence time of the photon pairs is fixed by the cavity linewidth. In our case $\tau_{pump} = 1 \mu s$ and $\tau_{pair}$ = 120 ns. By switching on the telecom detectors only during the time windows that we define as $|e\rangle$ and $|l\rangle$ we select the entangled state $|\Phi_{s,i}^+\rangle = 1/\sqrt{2}(|e_s e_i\rangle + |l_s l_i\rangle)$, with $s(i)$ representing the signal (idler) photon. More information about the energy-time entangled state generated with this source can be found in reference[27] and in the SI.

### Telecom qubits source

A successful BSM requires indistinguishability in all possible degrees of freedom between the qubit states generated by Bob and the idler photons. In order to generate weak coherent states at the same wavelength as the idler photons (1436 nm) we use a second non-linear crystal placed inside a resonator with the same specifications as that of the cSPDC source. We operate it as an optical parametric oscillator (OPO) and perform difference frequency generation (DFG) between the pump laser (426 nm) and the reference 606 nm light. We send this coherent laser light through optical attenuators that reduce its intensity to the single photon level. Finally, these weak coherent states are modulated in phase and amplitude to encode the time bin states used

as input qubits: $|\phi_{IQ}\rangle = \alpha|e_{IQ}\rangle + e^{i\varphi}\beta|l_{IQ}\rangle$. For all the experiments, we used a mean photon number per qubit of 0.02 at the beam splitter of the BSM. More information about the OPO and input qubits generation can be found in reference[18] and in the SI.

### Bell-state measurement

The joint state of $|\Phi_{s,i}^+\rangle$ and $|\phi_{IQ}\rangle$ can be re-written as:

$$
\begin{aligned}
|\Phi_{s,i}^+\rangle \otimes |\phi_{IQ}\rangle \propto &|\Phi_{i,IQ}^+\rangle(\alpha|e_s\rangle + e^{i\varphi}\beta|l_s\rangle) \\
&+ |\Phi_{i,IQ}^-\rangle(\alpha|e_s\rangle - e^{i\varphi}\beta|l_s\rangle) \\
&+ |\Psi_{i,IQ}^+\rangle(e^{i\varphi}\beta|e_s\rangle + \alpha|l_s\rangle) \\
&+ |\Psi_{i,IQ}^-\rangle(e^{i\varphi}\beta|e_s\rangle - \alpha|l_s\rangle)
\end{aligned}
\tag{1}
$$

with the Bell-states $|\Phi_{i,IQ}^\pm\rangle = 1/\sqrt{2}(|e_i e_{IQ}\rangle \pm |l_i l_{IQ}\rangle)$ and $|\Psi_{i,IQ}^\pm\rangle = 1/\sqrt{2}(|e_i l_{IQ}\rangle \pm |l_i e_{IQ}\rangle)$. Detecting two consecutive events in the same output of the BSM beam-splitter projects the joint state of the idler and input qubit fields onto $|\Psi_{i,IQ}^+\rangle$ and detecting two consecutive events in opposite outputs projects it onto $|\Psi_{i,IQ}^-\rangle$. Consequently, a $|\Psi_{i,IQ}^+\rangle$ event heralds a quantum teleportation event of the state $e^{i\varphi}\beta|e_s\rangle + \alpha|l_s\rangle$ to the signal photon and a $|\Psi_{i,IQ}^-\rangle$ event heralds the teleportation of the state $e^{i\varphi}\beta|e_s\rangle - \alpha|l_s\rangle$. In order to always have the same quantum state before the qubit analysers we have to apply a unitary transformation conditioned on the result of the BSM. Such transformation consists of a $\pi$-phase of the relative phase between the $|e\rangle$ and $|l\rangle$ time bins of the $|\Psi_{i,IQ}^-\rangle$ heralded teleported states.

For the short distance scenario, conditioned on a successful BSM, the probability to have a teleported qubit before (after) the QM is $7.1 \cdot 10^{-2}$ ($1.2 \cdot 10^{-2}$) while for the 1 km scenario it corresponds to $6.5 \cdot 10^{-2}$ ($7.5 \cdot 10^{-3}$). The small variations between the values measured before the QM can be explained due to day-to-day variations in the experimental set-up. We heralded the teleportation events with a BSM rate of the order of 1 Hz with small variations from day-to-day. More information about the BSM and how the unitary transformation is applied can be found in the SI.

### Computing the measured fidelities

As explained in the main text, for every input qubit ($|x\rangle$) we analysed in its parallel ($\langle x|$) and orthogonal ($\langle y|$) settings. From each measurement we computed the visibility ($V$). When the input qubit was on the equator of the Bloch-sphere, $V_{|x\rangle} = \frac{C_{\langle x|x\rangle} - C_{\langle y|x\rangle}}{C_{\langle x|x\rangle} + C_{\langle y|x\rangle}}$ where $C$ corresponds to the coincidences after teleporting the state $|x\rangle$ and analysing it with the settings $\langle x|$ or $\langle y|$. Instead, when teleporting a qubit on a pole of the Bloch sphere, due to the missing bit-flip transformation, the visibility was calculated as $V_{|x\rangle} = \frac{C_{\langle y|x\rangle} - C_{\langle x|x\rangle}}{C_{\langle y|x\rangle} + C_{\langle x|x\rangle}}$ (e.g. when teleporting the state $|e\rangle$ the state after the QM will be $|l\rangle$). Once we measured the visibilities of the different states we calculated the fidelities of the individual teleported states as $F_{|x\rangle} = \frac{1 + V_{|x\rangle}}{2}$ and finally we calculated the mean fidelity for an arbitrary qubit as $\bar{F} = \frac{1}{3}\bar{F}_{poles} + \frac{2}{3}\bar{F}_{eq}$[13]. Note that the values of fidelities reported in the text are not corrected for any background.

### Classical limit for fidelity

In the main text we compared the measured fidelities with the classical limit of 66.7 %. This bound is obtained by assuming a measure-and-prepare strategy with qubits encoded into single photons. A different strategy could exploit the Poissonian statistics of the weak coherent state qubit that is teleported to perform a similar measure-and-prepare strategy[37]. In this scenario, the classical limit depends on the mean photon number of the input qubit and on the efficiency of the quantum process that is replaced by the classical strategy. In our case, this is

given by the probability per BSM of heralding a photon after the QM, which was $1.2 \cdot 10^{-2}(7.5 \cdot 10^{-3})$ for the short distance (1 km distance) scenario. Using the mean photon number of 0.02, we find a maximum fidelity achievable with a classical strategy of 73.6 % (75 %). All our measured fidelities significantly violate all the classical bounds.

Finally, when the input qubit is sent in the equator of the Bloch-sphere, we can measure what is the fidelity of the state after the memory if we do not condition its analysis on a Bell-state measurement, i.e. if we consider all the single detection events at Bob. We obtained $\bar{F}_{eq} = 53.5(7)$ % and $\bar{F}_{eq} = 53(1)$ % for the 5 m and 1 km cases respectively, far below the measured fidelities.

## Data availability
The data that support the main findings of this study are available in the following public data repository[38].

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

## Acknowledgements

The authors thank S. Payne for his advice on the operation and alignment of the phase shifter and L. Mind for their help in monitoring of the experiment. This project received funding from the European Union Horizon 2020 research and innovation program within the Flagship on Quantum Technologies through grant 820445 (QIA) and under the Marie Skłodowska-Curie grant agreement No. 713729 (ICFOStepstone 2, for J.V.R.) and No. 754510 (proBIST, for S.G.), from the European Union Regional Development Fund within the framework of the ERDF Opera-tional Program of Catalonia 2014-2020 (Quantum CAT), from the Gor-don and Betty Moore Foundation through Grant GBMF7446 to HdR, from the Government of Spain (PID2019-106850RB-I00 (QRN), PLEC2021-007669 (Q-Networks) and Severo Ochoa CEX2019-000910-S, funded by MCIN/AEI/10.13039/501100011033), from MCIN with funding from

European Union NextGenerationEU (PRTR-C17.I1), from Fundació Cellex, Fundació Mir-Puig, and from Generalitat de Catalunya (CERCA, AGAUR).

## Author contributions

D.L.-R. and S.G. designed the experiment. The measurements were conducted by D.L.-R. and J.V.R., who also jointly analysed the data. D.L.-R. wrote the paper with input from all co-authors. H.d.R. supervised the project.

## Competing interests

The authors declare no competing interests.
