## [Peer Review File · Nature Communications]

Long distance multiplexed quantum teleportation from a telecom photon to a solid-state qubitREVIEWER COMMENTS

Reviewer #1 (Remarks to the Author):

This paper reports a teleportation experiment from a photonic qubit at telecom band to a quantum memory based on rare-earth (Pr:YSO) doped crystals. The authors combine the technology of cSPDC entanglement source and AFC memories they developed previously to experimentally demonstrate a quite complete conditional teleportation experiment, achieving high fidelity and high rep. rates. Compared with previous teleportation experiments involving similar setups, the key advance is the much longer storage time which allows the implementation of feedforward control on the teleported qubit, and the time-multiplexing of the experiment to enable higher teleportation rate than the single mode version of this protocol. The results are clearly presented with comprehensive description of the experimental details in the methods section and the SI. I believe the results represent an important development in quantum networking and quantum optics, and are worthy of publication. To help further enhance the clarity and readability, I have a few suggestions and questions:

1. The title of the paper mentions a teleportation of a photonic qubit to a solid-state qubit. While I understand what the authors intend to describe, I think the word "solid-state qubit" can be a bit misleading to general audience. To me, a solid-state qubit (or matter qubit) would imply a single particle (single atom, single ion etc) with two qubit encoding levels. It is typically perceived that way. In this work, the qubit is really a photonic qubit that is stored as collective excitation among the rare-earth ensembles. I would suggest changing to a more conventional description, such as "from a telecom photonic qubit to a solid-state memory". This would not impact at all the quality and impact of the work.
2. The "coincidence" histogram in Fig. 2b and the related text are a bit unclear to me. The coincidence means detection between two or more events that are taking place at the same time. In this experiment, depending on the input state (equator state or pole state), the "coincidence" measurement looks like they mean different things. While I can follow the description in the SI, it would be nice if there is a clear definition of the "coincidence" measurements they performed. If I understand correctly, for equator state inputs, coincidence means the successful BSM AND a photon detection after the analysis crystal? And for pole state inputs, coincidence means a successful BSM AND direct detection in either time bin after the QM? Please clarify.
3. I am curious why the etalon filter for the 606 nm photon has 4 GHz bandwidth, while the FP cavity filter for the telecom photons is of 80 MHz. Why not use similar bandwidth filters? And Why a fiber is used between the QM and the analysis crystal? Is it for mode cleaning or two set ups are too far apart?
4. In Fig. 2b, should the time-bin interval be 420 ns?
5. In Fig. 2b, how should I understand the coincidence outside the time window of interest (~420 ns)? There seems to be a lot of coincidence signals at the background, and it is not clear they represent the usual 3 peaks in time-bin interference pattern. Could the authors give more explanations?
6. One great feature of this experiment is the enhanced teleportation rep rate beyond the photon transit time limit. While the rep rate is certainly an important metric, what about other metrics such as efficiency i.e. what is the probability of a successfully teleported qubit (before and after analysis and detection) for each input qubit. It'd be helpful to specify this efficiency in the text. While I do not expect it, does this efficiency change with increasing rep. rate?
7. There are a couple of typos in the manuscript. For instance, pg. 2 last paragraph, it should be "to overcome the limit".

Reviewer #2 (Remarks to the Author):

The manuscript by Lago-Rivera et. al. describes an experimental realization of long distance quantum teleportation from a photonic qubit at telecom wavelength to a solid-state quantum memory separated by 1 km of optical fiber. They focus on multiplexed properties of the teleportation process thanks to

the multimode quantum memory they use. They also implemented an active feed-forward scheme performing a phase shift on the qubit retrieved from the memory to complete teleportation protocol. The resulting state shows fidelities of the retrieved state above classical threshold, enough to claim successful quantum teleportation. I believe that the results are correct and the experiment and analysis is sound and well done. The manuscript is well written and is of high quality.

I recommend this paper for publication in nature communications, but suggest authors to justify and better explain their claims.

- In their previous experiment [18], showing entanglement swapping between two quantum memories they also demonstrate multiplexed operation with 62 temporal modes. However, in the present paper there is no clear comparison with the previous work to explain better the novelty of multiplexed teleportation. In the present work they claim to increase the overall protocol speed by three times compared to single mode operation, but it is not clear how this relates to the multimode capacity and what is the limit.

- In the abstract, (and further in the main text) authors claim their 'active feed-forward scheme, implementing a phase shift on the qubit retrieved from the memory' completes the teleportation protocol. But later say that the 'additional bit flip required to recover the original qubit' after teleportation was not implemented. I suggest to clarify this to avoid any confusion.

Below I provide more detailed comments.

- were there any improvements in this experiment on any metric for the basic constituent processes (memory efficiency, heralding efficiency, storage time etc.) over the prior cited works by the same group?

- it is not clear if authors use background correction for their analysis. The effect from finite cross correlation is not explained;

- it is hard to extract detection rates from the presented data. In my opinion, a discussion on 'coincidence rate per teleportation event' is necessary to compare to other approaches;

- on page 5, line 117 the time-bin duration of 840ns is stated, instead of 420 ns used in the experiment;

REVIEWER COMMENTS

Reviewer #1 (Remarks to the Author):

This paper reports a teleportation experiment from a photonic qubit at telecom band to a quantum memory based on rare-earth (Pr:YSO) doped crystals. The authors combine the technology of cSPDC entanglement source and AFC memories they developed previously to experimentally demonstrate a quite complete conditional teleportation experiment, achieving high fidelity and high rep. rates. Compared with previous teleportation experiments involving similar setups, the key advance is the much longer storage time which allows the implementation of feedforward control on the teleported qubit, and the time-multiplexing of the experiment to enable higher teleportation rate than the single mode version of this protocol. The results are clearly presented with comprehensive description of the experimental details in the methods section and the SI. I believe the results represent an important development in quantum networking and quantum optics, and are worthy of publication. To help further enhance the clarity and readability, I have a few suggestions and questions:

We thank the reviewer for their comments on our work and their appreciation on the quality of our results and clarity of the manuscript.

1. The title of the paper mentions a teleportation of a photonic qubit to a solid-state qubit. While I understand what the authors intend to describe, I think the word "solid-state qubit" can be a bit misleading to general audience. To me, a solid-state qubit (or matter qubit) would imply a single particle (single atom, single ion etc) with two qubit encoding levels. It is typically perceived that way. In this work, the qubit is really a photonic qubit that is stored as collective excitation among the rare-earth ensembles. I would suggest changing to a more conventional description, such as "from a telecom photonic qubit to a solid-state memory". This would not impact at all the quality and impact of the work.

We appreciate the intention of the reviewer's comment but we would prefer to keep the current title. We believe that it is fair to call the matter system onto which we teleport a qubit. The two states of the time-bin qubit are preserved in the quantum memory, mapped onto two collective excitations which rephase at different times. While we understand that for some readers this could be an unfamiliar definition, we would like to remark that in the abstract we explicitly mention that our "matter qubit" consists of "... a collective excitation in a solid-state quantum memory".

2. The "coincidence" histogram in Fig. 2b and the related text are a bit unclear to me. The coincidence means detection between two or more events that are taking place at the same time. In this experiment, depending on the input state (equator state or pole state), the "coincidence" measurement looks like they mean different things. While I can follow the description in the SI, it would be nice if there is a clear definition of the "coincidence" measurements they performed. If I understand correctly, for equator state inputs, coincidence means the successful BSM AND a photon detection after the analysis crystal? And for pole state inputs, coincidence means a successful BSM AND direct detection in either time bin after the QM? Please clarify.

As we already state in the main text, a BSM always consists of two consecutive detections separated by 420 ns, regardless of the input state. As the reviewer points out, the only difference is with the analyser. Indeed, what we mean by coincidence corresponds always to a 3-fold detection (2 from the BSM and 1 from the teleported qubit). The reason to label the y-axis of Fig. 2b as "Coincidences

per teleportation event” is to renormalize the number of events to a common magnitude that does not depend on the measuring time.

To make this point clear, we modified the y-axis label of Fig. 2b as “3-fold coincidences per successful BSM” and in the caption we added: “3-fold coincidence histograms after parallel and orthogonal analysers, renormalized to the total amount of successful BSMs for an input state $|R\rangle$.”

3. I am curious why the etalon filter for the 606 nm photon has 4 GHz bandwidth, while the FP cavity filter for the telecom photons is of 80 MHz. Why not use similar bandwidth filters? And Why a fiber is used between the QM and the analysis crystal? Is it for mode cleaning or two set ups are too far apart?

The two filters are used for different purposes. The FP cavity is to filter out other frequency modes, as the spectrum of the source consists of around 14 frequency modes separated by 261 MHz. As the QM in this experiment can store only the central one, we need to ensure single frequency heralding to avoid uncorrelated detections in the idler path. For this reason, the filter in the telecom path has to be as narrow as 80 MHz, such that it can efficiently filter out the neighbouring frequency modes 261 MHz away. In the case of the 606 nm photon, the AFC stores the central frequency mode, while the inhomogeneous broadening of the Pr doped crystal, spanning a range of around 10 GHz, absorbs the extra frequency modes, such that the QM effectively acts as a narrow filter. The purpose of the etalon is to filter out noise outside of the absorption line of the QM crystal.

We now include this discussion in the SI, together with a richer bibliography with a better characterization of the spectrum of the source:

A more detailed discussion about the spectral characterization of the cSPDC source can be found in \cite{Rielander2016, Rielander2017, Seri2018}.

Finally, one of the reasons of having a fibre between the QM and the filter is indeed for spatial mode cleaning and also to be able to characterise each crystal independently if needed.

4. In Fig. 2b, should the time-bin interval be 420 ns?

The time-bin separation is indeed 420 ns. However, the time indicated in Fig. 2b (320 ns) corresponds to the coincidence window used in our analysis and it is related to the time duration of the photons, not to the separation between time bins. Note that an important condition to fulfil in our implementation is that the duration of the photon has to be smaller than the separation between the early and late bins to ensure orthogonality between the two possible states.

5. In Fig. 2b, how should I understand the coincidence outside the time window of interest (~420 ns)? There seems to be a lot of coincidence signals at the background, and it is not clear they represent the usual 3 peaks in time-bin interference pattern. Could the authors give more explanations?

As the reviewer correctly points out, in Fig. 2b it is possible to see coincidence counts outside of the desired window. They indeed represent the usual 3 peaks after a time-bin analyser due to its probabilistic nature, where only the central bin is sensitive to the phase setting of the interferometer, as can be appreciated by comparing upper and lower histograms. To avoid confusion, we added to the caption of Fig. 2b the following text: “b. ... Three coincidence peaks

appear in the analysis where only the central one corresponds to a projection in the superposition basis.”

6. One great feature of this experiment is the enhanced teleportation rep rate beyond the photon transit time limit. While the rep rate is certainly an important metric, what about other metrics such as efficiency i.e. what is the probability of a successfully teleported qubit (before and after analysis and detection) for each input qubit. It'd be helpful to specify this efficiency in the text. While I do not expect it, does this efficiency change with increasing rep. rate?

We agree with the reviewer that the probabilities that they remark are metrics of interest. To make it more clear and give a better understanding to the readers about the performance of our setup, in the *Methods* we added a sentence at the end of the “Bell-state measurement” section stating: “For the short distance scenario, conditioned on a successful BSM, the probability to have a teleported qubit before (after) the QM is $7.1e-02$ ($1.2e-02$) while for the 1 km scenario it corresponds to $6.5e-02$ ($7.5e-03$). The small variations between the values measured before the QM can be explained due to day-to-day variations in the experimental set-up. We heralded the teleportation events with a BSM rate in the order of 1 Hz with small variations from day-to-day.” Note that the efficiency of heralding a qubit after the QM for the short distance case corresponds to $1.2e-02$, and not $1.4e-02$ as we had mentioned in the previous version of the manuscript. We found this minor mistake in the previous analysis of the data. It has no influence in the claims of the article as the classical bound for the fidelity only goes from 72.7% to 73.6 % (we also updated this value in the revised manuscript).

Moreover, these values do not depend on the repetition rate of the experiment thanks to the temporal multimodality of the AFC, which absorbs consecutive photons / temporal modes with equal efficiency. The maximum repetition rate is then given by the duration of the time bin qubit, such that each bin has a spectrum matching the QM bandwidth and the overlap between the two bins is minimum.

7. There are a couple of typos in the manuscript. For instance, pg. 2 last paragraph, it should be "to overcome the limit".

Thanks for this remark, we modified it to what we actually meant: “...we were able to increase the repetition rate of our experiment by a factor of about three beyond the limit set for a single mode”.

We also revised the manuscript and found a few other typos that we corrected.

Reviewer #2 (Remarks to the Author):

The manuscript by Lago-Rivera et. al. describes an experimental realization of long distance quantum teleportation from a photonic qubit at telecom wavelength to a solid-state quantum memory separated by 1 km of optical fiber. They focus on multiplexed properties of the teleportation process thanks to the multimode quantum memory they use. They also implemented an active feed-forward scheme performing a phase shift on the qubit retrieved from the memory to complete teleportation protocol. The resulting state shows fidelities of the retrieved state above classical threshold, enough to claim successful quantum teleportation. I believe that the results are correct and the experiment and analysis is sound and well done. The manuscript is well written and is of high quality.

I recommend this paper for publication in nature communications, but suggest authors to justify and better explain their claims.

We thank the reviewer for their positive comments on our work and further remarks to improve the clarity of the manuscript.

- In their previous experiment [18], showing entanglement swapping between two quantum memories they also demonstrate multiplexed operation with 62 temporal modes. However, in the present paper there is no clear comparison with the previous work to explain better the novelty of multiplexed teleportation. In the present work they claim to increase the overall protocol speed by three times compared to single mode operation, but it is not clear how this relates to the multimode capacity and what is the limit.

The relation between the gain in repetition rate and the multimode capacity is the following. For a storage time of 10 us (as used in Fig. 2, Fig 3a. and 3b.) our memory can store 25 modes. Considering a communication time equal to the AFC storage time of 10 us (and therefore a separation of 1 km), we can achieve a repetition rate of 323 kHz with our multimode memory, while the maximum repetition rate is 100 kHz. However, as we mention in the manuscript, the maximum repetition rate is currently limited by our electronics. By using faster electronics, we could reach a repetition rate of 1.2 MHz, limited only by the duration of one qubit. This would lead to a gain by a factor 12. This factor corresponds to how many time-bin qubits can be stored in the quantum memory, equal to half the total number of temporal modes which can be stored. However, if we consider a different encoding which makes use of only one temporal mode (e.g. polarisation) then our memory would bring a gain of 25 with respect to a single mode equivalent.

There is also an important difference compared to our previous experiment in Ref [18]. In [18], the long distance was only simulated while in the present work we actually use a long fibre. This means that for this experiment the multimode capacity is used to increase the repetition rate in a practical scenario.

- In the abstract, (and further in the main text) authors claim their 'active feed-forward scheme, implementing a phase shift on the qubit retrieved from the memory' completes the teleportation protocol. But later say that the 'additional bit flip required to recover the original qubit' after teleportation was not implemented. I suggest to clarify this to avoid any confusion.

While applying the pi-phase shift as we do is enough to ensure that we always have the same teleported qubit before the analyser, we agree with the referee that the claim as it is can lead to confusion. Therefore, we modified this sentence in the abstract to be: Our system encompasses an active feed-forward scheme, implementing a conditional phase shift on the qubit retrieved from the memory, as required by the protocol (we also modified this claim in line 55 where the same sentence is repeated).

Below I provide more detailed comments.

- were there any improvements in this experiment on any metric for the basic constituent processes (memory efficiency, heralding efficiency, storage time etc.) over the prior cited works by the same group?

The performances of the source and memory are comparable to the ones reported in Ref [18]. The novel “enabling tools” that made possible this experiment were instead: to develop a source of time-bin qubits indistinguishable from the entangled telecom photons, to implement the BSM and develop a method to actively apply the pi-phase shift depending on its outcome.

- it is not clear if authors use background correction for their analysis. The effect from finite cross correlation is not explained;

We did not perform any background corrections for the measurements presented in this paper. We now explicitly mention it in the *Methods*.

Note that the last section in the SI discusses the limitations on the measured fidelities. In the subsection “C. Statistics of a weak coherent input qubit” we compute a model inspired by reference Valivarthi2016. This model implicitly accounts for the effect of multi-photon pairs and therefore finite cross-correlation on the fidelity. To make it explicit we modified the first paragraph of this section as: “In the Supplementary Information of [7], section 5, the authors introduce an analytical model to estimate the fidelity limitations depending on the mean photon number used in the input qubits and the probability of generating an entangled photon pair at their SPDC source.”

- it is hard to extract detection rates from the presented data. In my opinion, a discussion on 'coincidence rate per teleportation event' is necessary to compare to other approaches;

In line with a similar question from reviewer #1, we have now added a paragraph in the *Bell-state measurement* section of the Methods where we state the different heralding efficiencies of a BSM event in different parts of our experiment together with the BSM rate:

“For the short distance scenario, conditioned on a successful BSM, the probability to have a teleported qubit before (after) the QM is $7.1e-02$ ($1.2e-02$) while for the 1 km scenario it corresponds to $6.5e-02$ ($7.5e-03$). The small variations between the values measured before the QM can be explained due to day-to-day variations in the experimental set-up. We heralded the teleportation events with a BSM rate in the order of 1 Hz with small variations from day-to-day.”

- on page 5, line 117 the time-bin duration of 840ns is stated, instead of 420 ns used in the experiment;

The 840 ns that we wrote in line 117 is correct. The two temporal modes of the time-bin qubit are each 420 ns wide, and they are separated by 420 ns, amounting to a total length of 840 ns. We have modified the text of the “temporal multiplexing” section, adding additional explanation to avoid further possible confusion for the reader:

“If Alice and Bob are spatially separated, the teleportation repetition rate is limited by the two-way communication time between the two parties. For example, a separation of 1 km would correspond to a communication time of 10 μ s, equal to the storage time of the QM. Thanks to the temporal multimodality of our quantum memory, we can overcome this limitation. We repeated the experiment varying the teleportation repetition rate from 133 kHz to 323 kHz, using $|R\rangle$ as the teleported qubit (Fig. 3a). We observed a constant fidelity for all the explored repetition rates (Fig. 3b). Note that the maximum rate used (323 kHz) was restricted by the speed of the experimental control electronics. Implementing a faster controller would allow the system to reach a repetition rate up to 1.19 MHz, only limited by the total duration of the time-bin qubit of 840 ns. If instead only a single qubit could be stored, after 1 km of distance the two-way communication time between Alice and Bob would lead to a maximum repetition rate of 100 kHz (black vertical line of Fig.3b). The multimode operation then introduces a maximum gain of a factor of 12 with respect to a single mode architecture. This number corresponds to how many time-bin qubits can be stored in the quantum memory, equal to half the total number of temporal modes which can be stored (about 24 for this storage time). In combination with the telecom compatibility of our approach, the temporal

multimodality of our QM makes our system especially suitable to be used in a long-distance scenario.”

#Extra

In addition to the previous remarks, while reviewing the manuscript, we noticed that the classical bound for the fidelity calculated using the strategy from Ref [37] depends on the scenario that we compare it to. Since we use a different storage time in the QM for the 1 km scenario, the classical bound should also be slightly modified due to the lower probability of heralding the presence of a teleported qubit after the memory. We recalculated this bound to be 75% (still compatible with the claims of the article) and modified the text accordingly.

REVIEWERS' COMMENTS

Reviewer #1 (Remarks to the Author):

I appreciate author's reply and revision to my comments. They addressed my questions well. I support the publication in the current form.

Reviewer #2 (Remarks to the Author):

The authors have addressed raised comments. It is very helpful to get more details about this work from their reply. I support publication of this manuscript.